# Assessing anxiety, depression and insomnia symptoms among Ebola survivors in Africa: A meta-analysis

Jeremiah W. Acharibasam[1], Batholomew Chireh[2], Hayelom G. Menegesha[3]*

1 Department of Community Health and Epidemiology, University of Saskatchewan, Saskatoon, Saskatchewan, Canada, 2 Saskatchewan Health Quality Council, Saskatoon, Saskatchewan, Canada, 3 College of Medicine and Health Sciences, Adigrat University, Adigrat, Ethiopia

* hayetgeb@gmail.com

## Abstract

### Background

During health disaster events such as the current devastating havoc being inflicted on countries globally by the SARS-CoV-19 pandemic, mental health problems among survivors and frontline workers are likely concerns. However, during such health disaster events, stakeholders tend to give more precedence to the socio-economic and biomedical health consequences at the expense of mental health. Meanwhile, studies show that regardless of the kind of disaster/antecedent, all traumatic events trigger similar post-traumatic stress symptoms among survivors, families, and frontline workers. Thus, our study investigated the prevalence of anxiety, depression and insomnia symptoms among survivors of the 2014–2016 Ebola virus disease that plagued the West African sub-region.

### Methods

We systematically retrieved peer-reviewed articles published between 1970 and 2019 from seven electronic databases, including Google Scholar, MEDLINE, PsychInfo, PubMed, Scopus, Springer Link, Web of Science on Ebola and post-traumatic stress disorder symptoms. A comprehensive hand search complemented this literature search. Of the 87 articles retrieved, only 13 met the inclusion criteria for this meta-analysis.

### Results

After heterogeneity, influence, and publication bias analysis, our meta-analysis pooled proportion effects estimates showed a moderate to a high prevalence of anxiety (14%; 99% CI: 0.05–0.30), depression (15%; 99% CI: 0.11–0.21), and insomnia (22%; 99% CI: 0.13–0.36). Effect estimates ranging from (0.13; 99% CI: 0.05, 0.28) through to (0.11; 99% CI: 0.05–0.22), (0.15; 99% CI: 0.09–0.25) through to (0.13; 99% CI: 0.08–0.21) and (0.23; 99% CI: 0.11–0.41) to (0.23; 99% CI: 0.11–0.41) were respectively reported for anxiety, depression and insomnia symptoms. These findings suggest a significant amount of EVD survivors are struggling with anxiety, depression and insomnia symptoms.

**Data Availability Statement:** All relevant data are within the manuscript and its Supporting information files.

**Funding:** The author(s) received no specific funding for this work.

**Competing interests:** The authors have declared that no competing interests exist.

## Conclusion

Our study provided the first-ever meta-analysis evidence of anxiety, depression, and insomnia symptoms among EVD survivors, and suggest that the predominant biomedical health response to regional and global health disasters should be complemented with trauma-related mental health services.

## Introduction

The Ebola virus disease (EVD) has a long history in Africa, dating back to 1976, where 318 cases and 280 deaths were first recorded in the Democratic Republic of Congo [1]. Most recently, two more outbreaks were reported in the Democratic Republic of Congo (DRC). The first started in May 2018 at the Equateur region of DRC and lasted for two months with 58 cases, and 27 deaths reported [2]. The second DRC outbreak started in August 2018 at North Kivu region and is ongoing [3] with 3453 cases (3310 confirmed and 143 probable) as well as 2273 deaths and 1169 survivors reported as of 31st March 2020 [3]. However, the recent Ebola virus disease outbreak that shook the foundations of already fragile health care systems primarily occurred between 2014–2016 in three West African countries namely Sierra Leone, Liberia and Guinea [4]. Its unprecedented nature resulted in a substantial increase in human suffering and deaths. According to WHO, an estimated 28, 646 confirmed cases and 11, 322 deaths were recorded within that period [4]. Several factors were attributed to this large-scale spread of the EVD. Apart from the resources constraints and already challenged healthcare systems in these West African countries [5,6], sociopolitical and cultural factors unique to the region [7] contributed significantly to the spread of the EVD virus. Despite the unprecedented EVD outbreak and its associated mortality and morbidity rates, it also recorded the highest number of survivors in the history of Ebola outbreaks. An estimated 10,000 people in the three West African countries survived the epidemic, according to WHO estimates [8].

In addition, regardless of the antecedent, traumatic events have been shown to trigger adverse psychological processes and experiences [9]. Thus, whether war victims, soldiers, victims of war disasters, first responders, or patients, the resulting post-traumatic psychological impact of these events is indistinguishable [10–12]. Unfortunately, it seems direct survivors/patients of health disasters like Ebola tend to receive lesser public health and policy attention than the other disaster events. This trend is no surprise as war veterans or war crimes are often perceived positively as national heroes or victimized minority groups, respectively. In contrast, survivors of health disasters such as infectious epidemics and pandemics can often be tagged with social suspicion and negative stigmatization. Post EVD survivors do not only face the traumatic experience of suffering from the disease but also have to deal with community fear and social stigmatisation issues within a society devastated by the outbreak [13,14]. Earlier epidemiological studies have documented the link between mental health problems and infectious diseases outbreaks [15,16].

Also, symptoms of anxiety, depression, and insomnia are found to be the most predominant mental health issues reported among post-disaster survivors (war veterans, Ebola survivors etc.) during and after an outbreak event [12,17–19]. Therefore, adverse mental health outcomes such as experience with ill individuals, perceptions of threat, high levels of mortality, food and resource insecurity, stigma and discrimination, and intolerance of uncertainty could be expected during and after the Ebola epidemic in the affected African countries.

However, research has shown that mental health issues are mostly neglected in low- and middle-income countries [20], which is often accompanied by minimal allocation of resources to mental health care. For instance, in 2015, a World Bank report noted that as of the time of the Ebola outbreak, the number of mental health workers (including psychiatrists) in the local population was as low as 1 in 6 million in Sierra Leone and 1 in 25,000 in Liberia [19]. This evidence exemplifies the tremendous need for both additional resources and novel approaches in outreach and treatment of mental health burden in West Africa. Also, one of the major reasons for the devastating Ebola outbreak was the initial slow response from international health partners [21,22]. Even when the support finally comes, current Western post-disaster health aid to tackling African epidemics seem to prioritize the biomedical interventions to the neglect of mental health [21,22]. This predominantly biased biomedical public health approach to post-disaster interventions may lead to inequitable distribution of health resources in these affected regions. Also, international humanitarian donor agencies that prioritize funding for a biomedical response over a complementary mental health one may continue to feed into a persisting biomedical-first approach at the expense of alternatives that incorporate mental health needs of survivors [21,22].

Therefore, continuing to promote the biomedical approach may unintentionally further weaken developing countries' capacity to effectively handle health crisis post-disaster mental health care needs of survivors. Also, as victims of health crisis do not only have to deal with the stress and trauma of surviving the illness but also the community stigmatizing reaction to them for falling ill, contact tracing can be affected during critical health disasters. Therefore, the goal of this study is to draw attention to the health crisis-induced post-traumatic stress disorder symptoms among survivors pandemics and epidemics, especially those in poorer countries. Our specific aim is to provide a meta-analysis effects estimates of the prevalence of anxiety, depression and insomnia symptoms (per the definition of the DSM-5) among African EVD survivors.

Currently, the majority of existing reviews have been qualitative assessments of psychological and neuropsychological consequences, lived experiences and coping strategies of Ebola disease among survivors [23,24]. Also, there appears to be non-existing meta-analysis evidence on the prevalence of anxiety, depression and insomnia symptoms as mental health burden among Ebola survivors in the affected countries in Africa. Thus, this meta-analysis presents a quantitative estimate of the prevalence of the above symptoms among EVD survivors. Providing this evidence could support clinical and policy efforts to improve post-health crisis interventions and resources allocation to mitigate the mental health impacts for survivors, families, and frontline workers.

## Methods

### Search strategy

The Preferred Reporting Items guide guided this Meta-Analysis for Systematic and Meta-Analysis (PRISMA) (See Fig 1) [25]. A systematic search was conducted on seven popular electronic databases, including Google Scholar, MEDLINE, PsychInfo, PubMed, Scopus, Springer Link, Web of Science to retrieve all available relevant literature on Ebola and post-traumatic stress disorder symptoms. Furthermore, a follow-up comprehensive hand search was carried out to enable us to capture grey literature, including thesis and dissertation works. Our exhaustive and rigorous search of literature helped to reduce the risk of leaving out important studies that could affect the quality of our meta-analysis conclusions [26]. Also, to ensure replicability, the same key terms were used across all the databases, including "post-traumatic stress disorder*", "post-Ebola syndrome*", PTSD, Ebola, "Ebola Virus*", "Ebola disease*", "Ebola

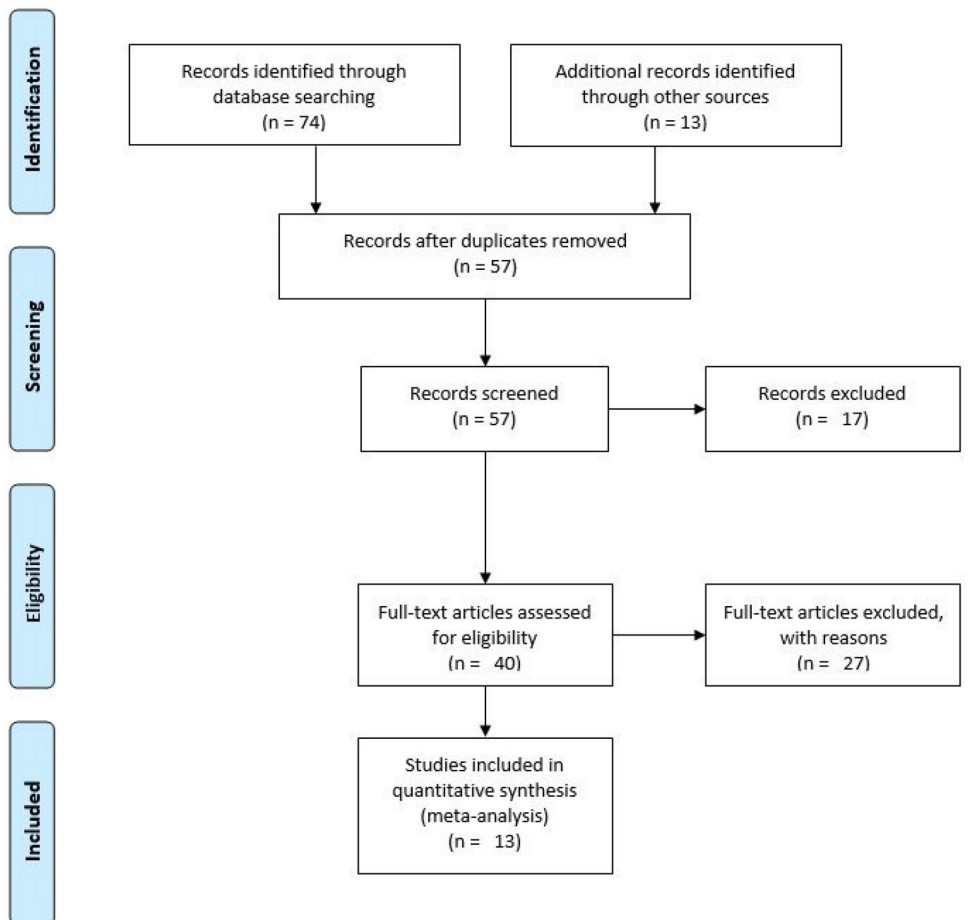

**Fig 1. PRISMA flow diagram-Ebola and anxiety, depression and insomnia symptoms.**

hemorrhagic fever", survivor, "Ebola survivor*", Africa*. We identified these key terms from relevant literature and MeSH terms in the various databases. We performed the searches on 14 January 2020. Please, find a sample of one of our search queries in S1 File. Ethical approval for this study was not necessary since it was a re-analysis of already published literature that can be accessed publicly.

## Inclusion/Exclusion criteria

The following inclusion/exclusion criteria were used for articles selection: 1) published between January 1, 1970, and December 31, 2019; 2) written in the English Language; 3) original and peer-reviewed; 4) cohort, cross-sectional, case-control and 5) from any African country. We did not include a specific age restriction for participants studied. However, we restricted our definition of anxiety, depression and insomnia to the one given by the DSM-5 [17]. Thus, we focused on studies that included any of the specified symptoms of anxiety, depression and insomnia per the DSM v5 definition due to that there was a paucity of existing literature. Another caveat was that only studies that provided raw frequency or proportion data on any of the known anxiety, depression and insomnia symptoms [17] and demographic information among Ebola survivors were included. This restriction was added to allow us to conduct proportional effect size meta-analysis and diagnostic analysis on the included studies.

For studies meeting the inclusion criteria but for which data could not be obtained in the article or the journal supplementary files' database, we contacted the authors via email to obtain raw data. Only a few of the authors contacted responded, and a couple still had the relevant data. Thus, we decided to restrict our analysis to only studies with complete data.

We excluded studies that were: 1) focused on Ebola and other non-mental health symptoms (anxiety, depression and insomnia); 2) done on non-human subjects; 3) conducted outside of Africa; 4) review studies, news articles, commentaries, letters and editorials, and 5) published before 1970. Studies that lacked raw frequency data but provided other results, including odds ratios, were discarded.

## Search results

We retrieved a total of 74 articles from the databases searched (Google scholar N = 29; Scopus N = 24; PubMed N = 8; Web of Science N = 7; Ovid Medline N = 3; Springer Link N = 2; Ovid PsycInfo N = 1). Also, the rigorous hand search returned 13 additional relevant articles on Ebola and symptoms of anxiety, depression and insomnia. Thus, a total of 87 electronic articles were obtained from the systematic literature search. From the 87 articles, 30 duplicates were discarded, and an additional 17 were removed after screening their titles and abstracts as they did not meet the inclusion criteria. Afterwards, 27 of the remaining 40 studies that failed to meet further inclusion limitations were eliminated after a full-text reading. Therefore, a remaining total of 13 articles were included in the meta-analysis (refer to Table 1 for a list of the added articles).

## Data collection and extraction

Author HGM was involved in the identification of the databases to be searched while authors J.W.A was responsible for the retrieval of articles from the electronic databases and author B.

Table 1. Summary of studies' attributes.

| First Author | Year | Setting | Study Design | Sample Size | Age (mean/ median) | Follow–up (years) | Assessment of Health Outcome |
|---|---|---|---|---|---|---|---|
| Clark et al. [40] | 2015 | Uganda | Retrospective Cohort | 49 | 40 (mean) | 5 months | Standardized Questionnaires |
| Maurice et al. [41] | 2018 | Liberia | Retrospective Cohort | 329 | 33(mean) | 10 months | Medical reports |
| Etard et al. [42] | 2017 | Guinea | Cross-sectional | 802 | 28.4 (median) | 1 year, 4 months | Standardized Questionnaires and clinical assessment |
| Wadoum et al. [43] | 2017 | Sierra Leone | Prospective Cohort | 246 | 27 (mean) | 1 year, 3 days | Questionnaires and clinical laboratory assessment |
| Howlett et al. [44] | 2018 | Sierra Leone | Prospective Cohort | 324 | 28 (median) | 4 months | Medical reports |
| Keita et al. [45] | 2017 | Guinea | Prospective Cohort | 256 | 31 (median) | 1 year | Standardized Questionnaires |
| Kelly et al. [46] | 2018 | Congo | Cross-sectional | 221 | 53.2 (mean) | 2 months | Standardized Questionnaires |
| Mohammed et al. [47] | 2017 | Sierra Leone | Retrospective Cohort | 115 | 28 (median) | 1 year, 1 month | Medical reports |
| Mohammed et al. [48] | 2015 | Nigeria | Cross-sectional | 117 | 34 (mean) | 2 weeks | Standardized Questionnaires |
| Qureshi et al. [49] | 2015 | Guinea | Cross-sectional | 105 | 38.9 (mean) | 3 months | Standardized Questionnaires |
| Scott et al. [50] | 2016 | Sierra Leone | Cross-sectional | 44 | 35 (median) | 3 weeks | Medical reports |
| Tiffany et al. [51] | 2016 | Sierra Leone | Retrospective Cohort | 166 | 24.7 (mean) | 5 months | Standardized Questionnaires and clinical assessment |
| Wilson et al. [52] | 2016 | Liberia | Cross-sectional | 268 | 30 (median) | 4 months | Standardized Questionnaires |

C. conducted the rigorous hand search for additional articles. Also, both authors independently examined all the 87 articles retrieved based on the above-stated eligibility criteria. Afterwards, 13 studies were included with information from independent samples on the following variables: country, design, sample, sex, age, follow-up period, stigma, anxiety, depression, insomnia. All the identified 13 studies presented raw frequency data enough for calculating summary statistics and proportional meta-analysis effects estimates with related 99% confidence intervals. An alpha level of 99% was used to ensure that the values of tau ($T$), tau-squared ($T^2$), and between-study variance ($I^2$) fall within the confidence intervals. For studies that met the inclusion criteria but did not have the required raw frequency data on the foregoing variables, Author J.W.A contacted authors via email to obtain this data. However, all articles providing insufficient data with no response from the contacted authors or failing to meet the inclusion criteria were ultimately discarded. Refer to Table 1 for details of the included studies and the related extracted. After retrieving all relevant research articles, author HGM was responsible for assuring the integrity of all data extracted.

## Data analysis and synthesis

For statistical analysis convenience, we stratified the 13 articles based on the sufficiency of frequency data provided by each study on the examined anxiety, depression and insomnia symptoms. These articles were classified into three meta-analysis groupings: (1) Ebola and anxiety; (2) Ebola and depression; (3) Ebola and insomnia. The studies in these three categories provided enough raw data to allow us to calculate summary proportion statistics and pooled meta-analysis effect sizes with their associated 99% confidence intervals using the metaphor [27] and meta [28] R programming language packages (version 3.6.3) [29]. All summary proportion statistics and pooled effects, sampling variances, and 99% confidence intervals estimates were computed using the inverse variance method based on the logit transformation (PLO) measure and restricted maximum-likelihood estimator (REML) random-effects method.

We assessed heterogeneity across studies in each stratum using the Q-statistic $X^2$ [30] and the DerSimonian and Laird $I^2$ statistic [31,32]. Further posthoc sensitivity analysis was conducted to identify influential studies (i.e. outlier effect sizes) that accounted for the heterogeneity in the meta-analysis data by using the studentized residuals (i.e. z-values) with a cut-off value set at 2 [33]. A cut-off of 2 was chosen due to the small number (i.e. <25) of studies within each stratum (ibid). We then used the Leave-one-out diagnostic test [34] to further confirm the effect of each study on the overall mean and estimate for the observed summary proportion by removing each study in a step-wise manner (ibid). Since the studies in each stratum were small, we considered the outlier effects insignificant and retained the studies in the meta-analysis if the number of outliers was less than two studies.

After the outlier analysis, we proceed to assess for risk of publication bias within each analysis stratum using funnel plots [35] based on the standard errors and the Eagger test [36] because it performs comparatively better when the meta-analysis involves a small number of studies (i.e. <25). Specifically, we used Eager's unweighted regression test because the traditional test is critiqued to have no theoretical justification [37]. However, we were unable to conduct subgroup/moderator and meta-regression analysis to account for the heterogeneity in effect sizes across the included studies because of the insufficiency of the data provided and the small number of studies under each analysis stratum [38,39]. Once all posthoc heterogeneity, diagnostic, and publication bias assessments were completed, we recalculated the meta-analysis summary proportion statistics and effect sizes and produced a final forest plot based on the precision of each study's effect size.

## Results

A total of 13 remaining articles were included in the meta-analysis [40–52]. Table 1 shows a detailed summary of the study attributes and data on the characteristics of the reviewed articles. Data were extracted based on the following features: country, design, sample, sex, age, follow-up period, presence of anxiety, depression, and insomnia. We considered the quality of the studies reviewed as high as there was no observed evidence of any publication bias.

### Ebola and anxiety

Five studies [41,44,47,48,52] assessed the prevalence of anxiety among Ebola survivors in this study. Anxiety was assessed via medical reports and standardized questionnaires. Fig 2a shows the individual study and pooled proportion effect estimates. The pooled prevalence of anxiety in these studies was 0.14 (99% CI: 0.05–0.30; $\chi^2$ = 45.91; $I^2$ = 94.63%; $p < 0.01$) among the EVD survivors, implying a significant level of anxiety. The absence of publication bias strengthens this finding (see Fig 2c). Most of the studies fell in the funnel plot within the 99% confidence interval and an Egger's [29] unweighted regression test ($p = 0.12$) showed a lack of asymmetry. Also, a sensitivity analysis using the Leave-one-out technique [27] produced effect estimates ranging from 0.13 (99% CI: 0.05, 0.28) through to 0.11 (99% CI: 0.05–0.22), showing a minimal influence of any individual study on the overall proportional effect estimate (see Fig 2b). Thus, a significant prevalence of anxiety is found.

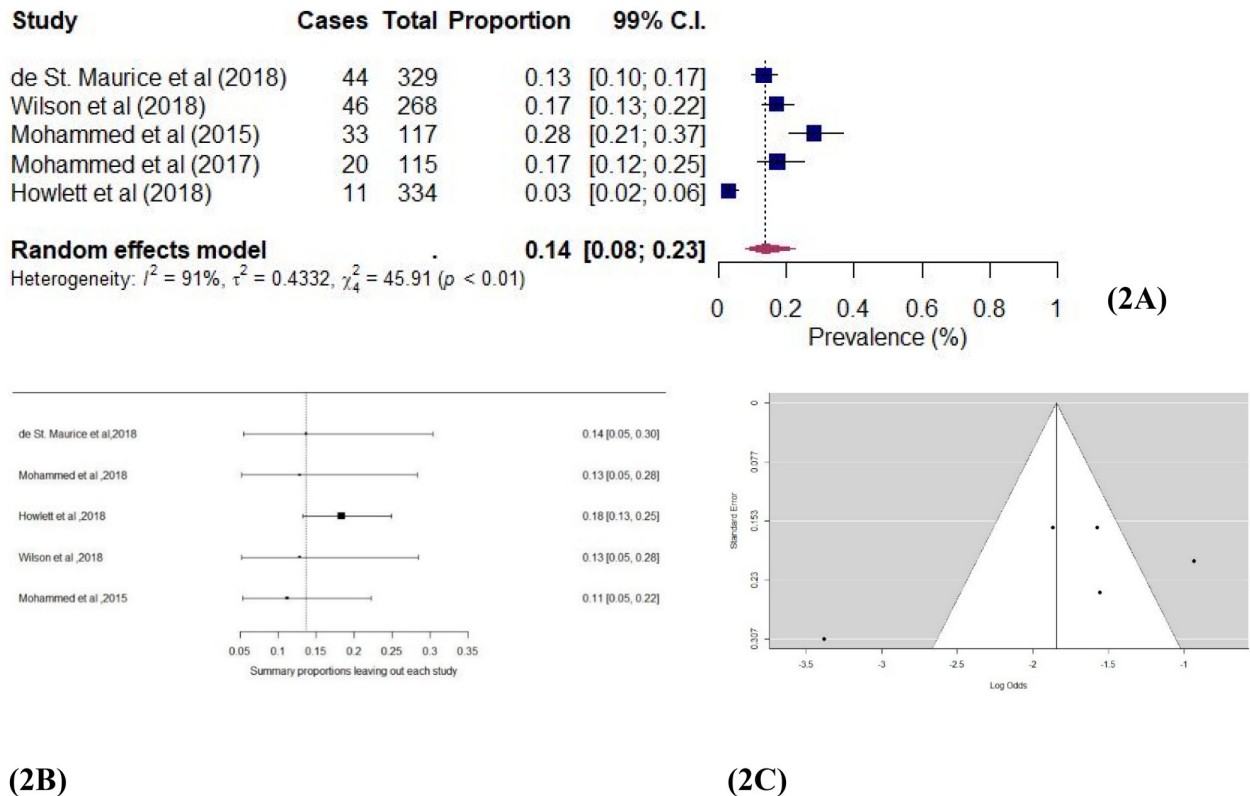

**Fig 2.** a Forest plot of Ebola and anxiety studies using the random-effects model. b: Sensitivity plot of Ebola and anxiety studies. c: Funnel plot of Ebola and anxiety studies.

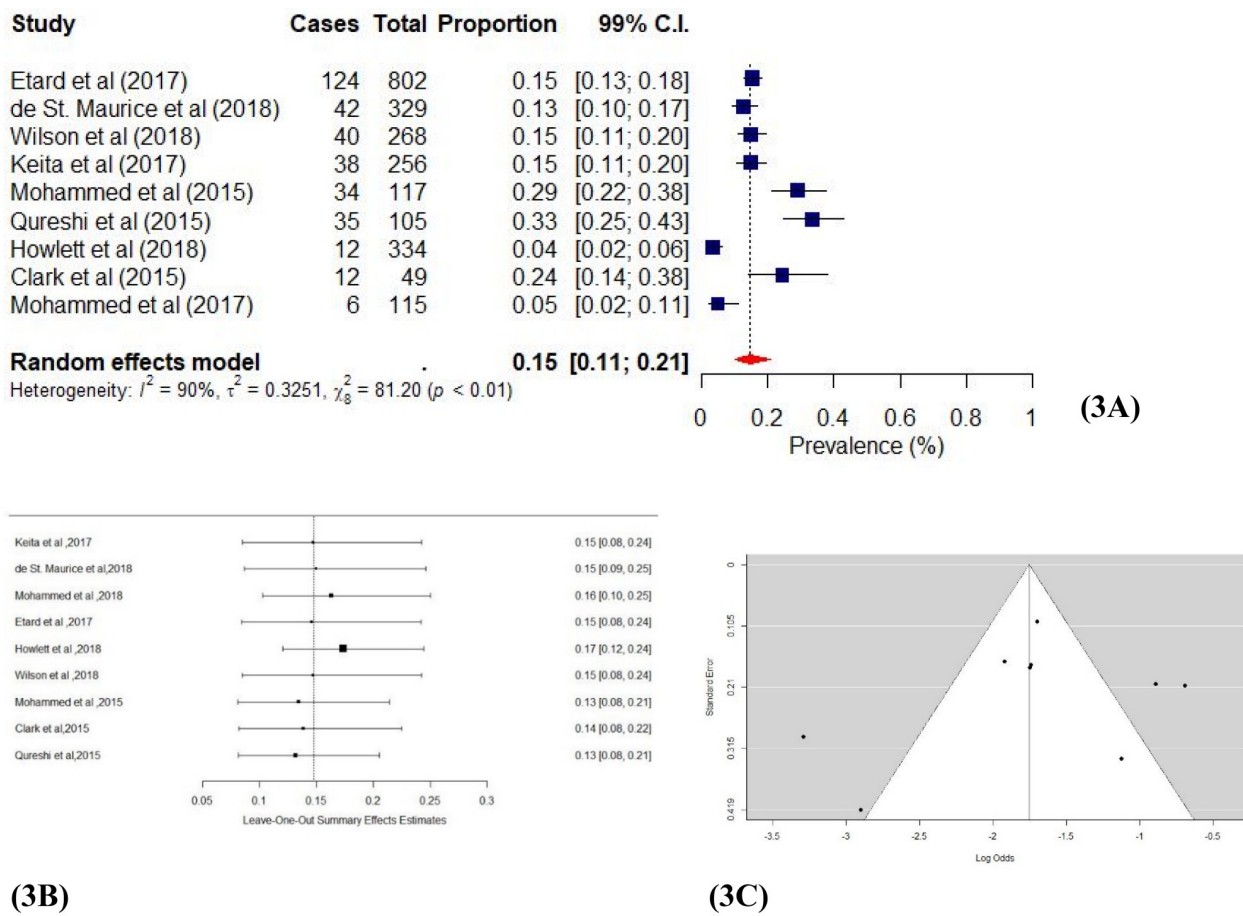

**Fig 3.** a Forest plot of Ebola and depression studies using random-effects model. b: Sensitivity plot of Ebola and depression studies. c: Funnel plot of Ebola and depression studies.

## Ebola and depression

Nine studies [40–42,44,45,47–49,52] estimated the prevalence of depression and/or depressive symptoms among Ebola survivors. Depression and/or depressive symptoms were assessed through medical reports, standardized questionnaires and clinical assessment. The results in Fig 3a shows the individual study and pooled proportion effect estimates. A pooled prevalence effect estimates of 0.15 (99% CI: 0.11–0.21; $\chi^2$ = 81.20; $I^2$ = 94.62%; $p < 0.01$) was reported, indicating a significant level of depression among EVD survivors. Publication bias assessments showed that half of the studies fell outside the funnel plots (Fig 3c), but Egger's [36] regression test indicated the absence of asymmetry ($p = 0.23$). Thus, the sensitivity results (Fig 3b) generally showed the effects estimates of almost all the studies distributed evenly around the reference line. Our leave-one-out analysis [34] supported these results with effect estimates ranging from 0.15 (99% CI: 0.09–0.25) through to 0.13 (99% CI: 0.08–0.21), which shows the minimal impact of individual studies on the main proportion effect estimate.

## Ebola and insomnia

Ten studies [40,41,43,44,47–52] where insomnia was assessed through medical reports and standardized questionnaires. Fig 4a shows the individual study and pooled proportion effect estimates. We found a pooled proportion effect estimate of 0.22 (99% CI: 0.13–0.36;

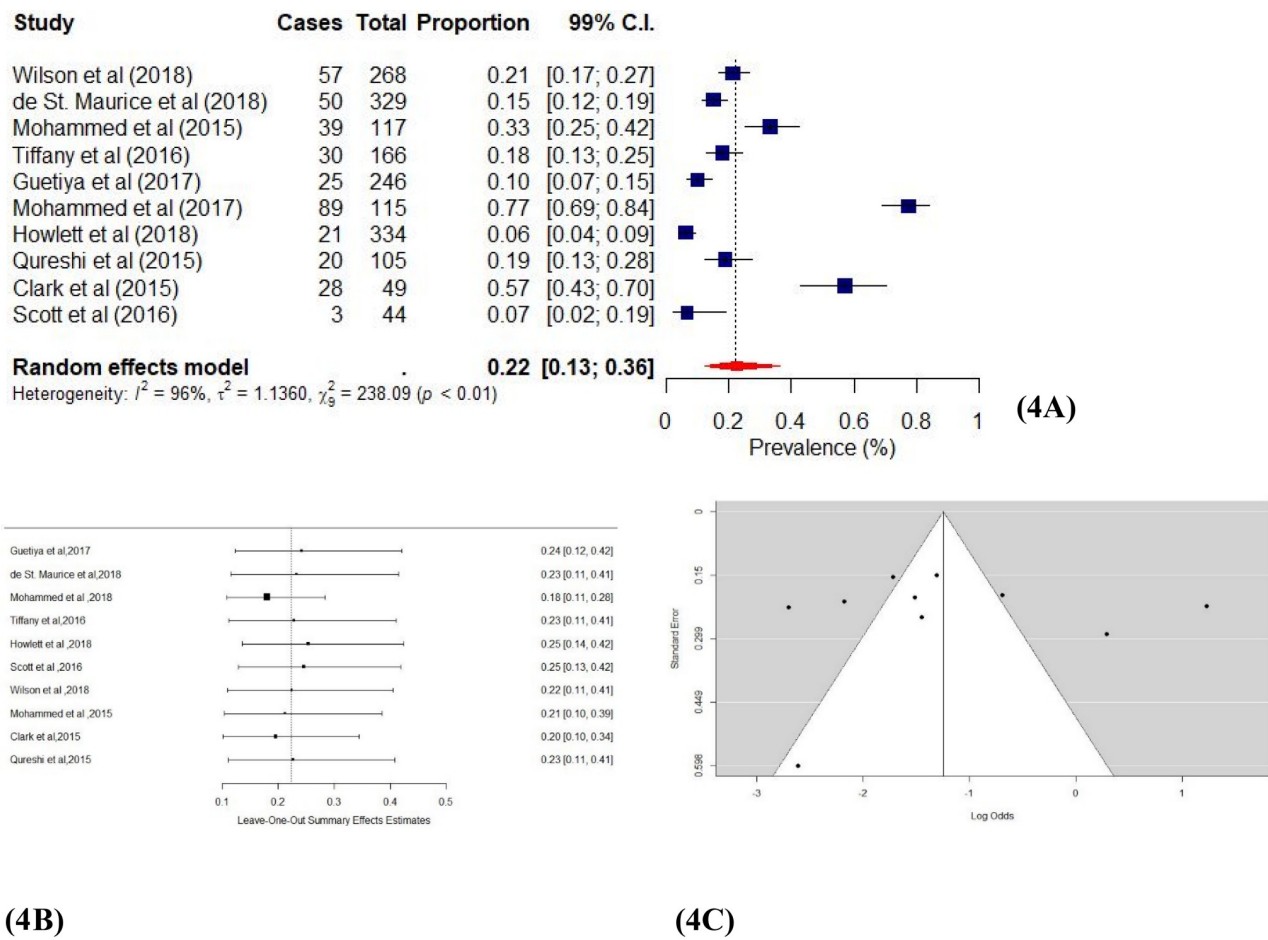

**Fig 4.** a Forest plot of Ebola and insomnia studies using the random-effects model. b: Sensitivity plot of Ebola and insomnia studies. c: Funnel plot of Ebola and insomnia studies.

$\chi^2 = 238.09$; $I^2 = 94.62\%$; $p < 0.01$), which indicates a significant presence of insomnia among EVD survivors. The funnel plot (Fig 4c) shows that most of the studies fall in the 99% confidence interval, and Egger's [29] regression test confirmed the absence of asymmetry ($p = 0.51$). The leave-one-out test [34] also showed the effect estimates ranging from 0.23 (99% CI: 0.11–0.41) to 0.23 (99% CI: 0.11–0.41) as each study is removed from the analysis. Thus, our sensitivity analysis showed a minimal influence of individual studies on the overall proportion effect estimate (Fig 4c). These results show a significant proportion of insomnia among EVD survivors.

## Discussion

The objective of this study was to provide pooled prevalence estimates of some common mental health problems amid health disaster events in West-Africa. Generally, our meta-analysis revealed findings regarding the presence of symptoms of anxiety, depression, and insomnia among Ebola virus disease survivors. Of the 13 studies involving 3042 Ebola virus disease survivors, a significant percentage of them reported anxiety, depression and insomnia symptoms. We found an estimated anxiety prevalence of 14% (99% CI: 0.05–0.30) among EVD survivors. Also, a significant proportion, 15% (99% CI: 0.11–0.21) of the survivors reported depression.

However, there was a higher prevalence of insomnia, 22% (99% CI: 0.13–0.36) among EVD survivors. Our results are slightly higher than what was reported in a recently published three-country study that found a 10.7%, 9.9% and 4.2% prevalence in anxiety among Ebola survivors in Sierra Leone, Liberia and Guinea respectively [53]. It is interesting to note that, studies that are restricted to only Ebola survivors tend to report lower estimates of anxiety, depression and insomnia compared to those conducted among the general population. In contrast to what we found, previous studies that reported on the mental health consequence of post-disaster occurrences in Sierra Leone showed a higher anxiety prevalence of 48% (95% CI: 46.8% to 50.0%) among the general population [54,55]. Although the reasons for the low prevalence estimates in these symptoms specifically among Ebola survivors are unclear, at the population level, researchers attributed the higher prevalence of people with mental health problems to region of residence, experiences with Ebola and perceived Ebola threat or knowing someone quarantined for Ebola [54,55].

Also, the findings of other studies support that there is a high prevalence of depression and reduced quality of life among survivors of EVD [23,56]. However, in contrast to the global estimate of 21% of depression among EVD survivors [23], our study only revealed a 15% pooled prevalence. We believe that the difference in proportions estimated could be due to the fewer number of studies used and our restriction to only studies in the African region. Similarly, with regards to insomnia, our prevalence estimates of 22% are lesser than a global estimate of 34% among EVD survivors [23]. We also attribute the differences in estimates to our regional restriction and fewer number of studies used in this meta-analysis. Taking the preceding into perspective, we believe these findings should be interpreted and applied with the needed caution.

Several studies have shown similarities between mental health outcomes among survivors of infectious disease outbreaks (e.g. the 2003 severe acute respiratory syndrome (SARS); 2009 novel influenza A (H1N1) pandemic; human immunodeficiency viruses (HIV/AIDS)) and other types of natural disasters including mass conflicts and displacement of people [57–59]. Accordingly, our study reveals, survivors of regional and global health crises such as EVD survivors are at an increased risk of symptoms of anxiety, depression, and insomnia. There is thus the need to scale up mental health services as a joint effort in combination with the normal biomedical health responses when faced with health disasters. Such coordinated efforts will demand comprehensive collaboration between governments, health professionals, donors, civil society, communities, and patients and their families to cater for vulnerable populations such as EVD survivors as recommended by the WHO Mental Health Gap Action Programme [60].

## Strengths and limitations of the study

A significant strength of this study is that, to the best of the authors' knowledge, this review is the first meta-analysis review of its kind to have estimated the prevalence of anxiety, depression and insomnia symptoms among EVD survivors in West Africa. Also, the relatively large sample size for our pooled estimates provides some significant power to our prevalence effect estimates. However, we acknowledge some limitations of our study.

First, although the recent outbreak has provided many new insights and a lot of new data, methodologically sound studies are still scarce. For instance, a majority of the studies used a cross-sectional design which prevents the establishment of a cause-effect relationship between EVD and anxiety, depression and insomnia symptoms. Thus, the heterogeneity in our proportion estimates of anxiety, depression and insomnia symptoms may simply be due to differences in the study designs, data quality, and/or demographic variations in various study samples.

Also, the higher number of cross-sectional studies affects our ability to draw cause-effect conclusions from our results. Secondly, as indicated, most of the articles we assessed did not compare EVD survivors to a control group. Therefore, we could not calculate measures of association (relative risks, odds ratios, etc.) between EVD and anxiety, depression and insomnia among survivors. Unfortunately, we contacted the authors but were unable to obtain the required data from the respective authors of the various studies assessed. Thus, we report our findings tentatively and advise readers to do likewise. We recommend that studies with more robust designs (e.g. cohort studies and case-control studies) should be conducted to establish the association between EVD and anxiety, depression and insomnia. Thirdly, there were no data on the severity of Ebola illness in all the papers assessed. Thus, our study could not distinguish between individuals who have these symptoms from others who do not. Finally, there were also differences in whether symptoms were based on clinical diagnosis or self-report. The mode of assessment of EVD patients' mental health symptoms was not uniform. While some studies reported the use of a standardized questionnaire, others reported medical reports as well as clinical assessment. The differences in measurement instruments and concerns about the quality of data prevent comparability of the studies we assessed.

## Conclusion

Findings from our review show that anxiety, depression and insomnia symptoms seem to be prevalent and widespread among EVD survivors. By highlighting the prevalence of mental health needs among EVD survivors, we hope this evidence supports policy and clinical efforts in allocating appropriate resources to mitigate the mental health impacts among survivors, families, and frontline workers during health disaster events (e.g. EVD and the current SARS-CoV-19 pandemic). We believe that providing program interventions, including organized mental health outreach activities, will go a long way to help EVD survivors navigate anxiety, depression and insomnia symptoms and improve their quality of life. However, similar to current mental health intervention efforts for veterans, a comprehensive and collaborative effort between relevant stakeholders is needed to achieve a successful complementary biomedical and mental health approach to health crisis survivors' in the future. Currently, few robust longitudinal studies examine the links between EVD and anxiety, depression and insomnia among survivors especially in Africa. Thus, we recommend robust longitudinal studies, including cohort and case-control studies, to help establish causal associations between EVD and anxiety, depression and insomnia as well as post traumatic stress disorder (PTSD).

## Supporting information

**S1 Checklist. PRISMA 2009 checklist.**
(DOC)

**S1 File.**
(XLSX)

## Acknowledgments

We acknowledge the contribution of earlier researchers and Ebola virus disease survivors for producing the data in this less researched area that we pooled and re-analysed in our present study.

## Author Contributions

**Conceptualization:** Jeremiah W. Acharibasam, Batholomew Chireh.

**Formal analysis:** Jeremiah W. Acharibasam, Batholomew Chireh.

**Methodology:** Jeremiah W. Acharibasam, Batholomew Chireh.

**Resources:** Hayelom G. Menegesha.

**Software:** Jeremiah W. Acharibasam, Batholomew Chireh.

**Supervision:** Jeremiah W. Acharibasam, Hayelom G. Menegesha.

**Validation:** Jeremiah W. Acharibasam, Batholomew Chireh, Hayelom G. Menegesha.

**Visualization:** Batholomew Chireh.

**Writing – original draft:** Jeremiah W. Acharibasam.

**Writing – review & editing:** Batholomew Chireh, Hayelom G. Menegesha.

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
