## [Decision Letter · Decision Letter 0]

15 Jul 2020

PONE-D-20-14937

Prevalence of posttraumatic stress disorder (PTSD) symptoms among Ebola survivors in Africa: a meta-analysis

PLOS ONE

Dear Dr. Mengesha,

Thank you for submitting your manuscript to PLOS ONE. After careful consideration, we feel that it has merit but does not fully meet PLOS ONE’s publication criteria as it currently stands. Therefore, we invite you to submit a revised version of the manuscript that addresses the points raised during the review process.

We look forward to receiving your revised manuscript.

Kind regards,

John Schieffelin, MD

Academic Editor

PLOS ONE

Journal Requirements:

Reviewers' comments:

Reviewer's Responses to Questions

**Comments to the Author**

1. Is the manuscript technically sound, and do the data support the conclusions?

Reviewer #1: Partly

Reviewer #2: Yes

2. Has the statistical analysis been performed appropriately and rigorously? 

Reviewer #1: Yes

Reviewer #2: I Don't Know

3. Have the authors made all data underlying the findings in their manuscript fully available?

Reviewer #1: Yes

Reviewer #2: Yes

4. Is the manuscript presented in an intelligible fashion and written in standard English?

Reviewer #1: Yes

Reviewer #2: Yes

5. Review Comments to the Author

Reviewer #1: This meta-analysis investigated the prevalence of PTSD symptoms among survivors of the 2014–2016 Ebola virus disease in the West African sub-region. Documenting the psychological impact of this mass public health crisis is important in order to inform both prevention and mitigation efforts as new crises emerge. A summary of studies of Ebola survivors is timely but several concerns need to be addressed for this paper to be a worthwhile contribution.

First, this should not be titled and described as a study of PTSD symptoms. Anxiety and depression are not specific PTSD symptoms, although some PTSD symptoms are similar to them (e.g.,. hyperarousal, negative mood). PTSD may involve insomnia but not all insomnia is trauma-related. It may be that experiencing Ebol;a was traumatizing, but without evidence linking the symptoms to Ebola-related experiences these cannot be assumed to be PTSD, they may be due to other stressors or pre-existing conditions, for example. The study should be described as assessing anxiety, depression, and insomnia symptoms, and the possible link to traumatization can be noted but not assumed. The lengthy and very basic description of PTSD should be deleted or greatly shortened.

Second, how does this study add new information beyond that reported in other reviews? Is it the first to bring together these or similar studies? IF not, what new information does it provide other than showing that prevalence estimates of the target symptoms are generally consistent across nationalities?

The absence of any comparison to estimates of the target conditions is understandable given that most of the studies did not include non-Ebola control groups. However, whether the prevalence estimates are clinically significant needs to be justified by contrasting the pooled estimates with prevalence data from community epidemiological studies. The prevalence of estimates for symptoms of anxiety (14%; 99% CI: 0.05–0.30), depression (15%; 99% CI: 0.11–0.21), and insomnia (22%; 99% CI: 0.13–0.36) are not clearly higher than estimates from other population surveys, making the authors' conclusion that "a significant amount of EVD survivors are struggling with common PTSD symptoms" unjustified.

Additionally, information is needed about how each of the conditions was operationalized in the various studies. Some stuidies used questionnaires: how were "anxiety" "depression" and "insomnia" determined to be present and clinically significant in the different studies? If symptom levels with a cutoff was used, was this based on relevant validation evidence? Other studies relied on medical reports: what was reported (diagnoses? impressions?) and what evidence was provided that the report yielded a valid classification of clinically significant symptoms?

The authors state that there was no publication bias, but how they determined this needs to be explained carefully.

Also, a major possible bias is the selection of participants. How did the studies demonstrate that their samples were representative of the larger group of Ebola survivors in their setting? If the samples are not clearly representative the findings are of limited value.

Were there no data on the severity of Ebola illness? This may distinguish individuals who have these symptoms from others who do not. If this was not reported it should be noted as a limitation.

Was gender not reported? Comparing men vs. women would add to the usefulness of the results.

Statements that describe the study or its findings in congratulatory manner should be deleted (e.g., "exciting" findings).

Reviewer #2: This study presents the results of a meta-analysis of PTSD symptoms among EVD survivors in sub-Saharan Africa. This study is indeed timely given the current COVID pandemic, and as the authors note, a lack of attention in both science and practice on the mental health sequelae of infectious diseases. Overall the article is well organized and well written, and I believe is poised to make an important contribution to the literature. I have a number of suggestions to improve the article, listed below.

Abstract

I would avoid beginning the abstract with a statement of the study’s timeliness. It would be better to lead with, “During health disaster events….”

I am not that familiar with methods in meta analyses, but I am accustomed to seeing effect sizes reported. In addition to prevalence estimates, is it possible to report effect sizes?

Introduction

The literature reviewed in this section is thorough and highlights the problems of mental health care particularly in LMICs. The section could be better organized to make it easier for the reader to digest, either through (a) more effective use of introductory sentences in each paragraph, or (b) the use of headings and subheadings.

The word “unfortunately” is used too much – in particular, see the beginning sentences on p. 10 and p. 11

Methods

This section is clear and succinct. For study inclusion and exclusion criteria, it would be helpful to know the number of studies included and excluded per criterion – if possible, within the flow chart depicted on p. 13.

Results

The information provided in the first paragraph is redundant with information already provided in the Methods section.

Discussion

The first paragraph basically repeats earlier information – I would advise starting this section with the main contribution of this study to existing literature.

On p. 22, replace with word “exciting” with a word more appropriate to the situation: “Generally, our meta-analysis revealed exciting findings….”

Also, on p. 22, the authors report a total of 3,042 EVD survivors included in the study. Are the authors sure that the various studies all used different samples? Or could there be overlap across studies in terms of participants?

In the Discussion section, please give some thought as to potential mechanisms behind the various prevalence rates presented. For example, why might insomnia have a higher prevalence than anxiety and depression? Also, the prevalence rates in the current study are much lower across the board compared to individual studies. Please go into some detail as to why the meta-analysis shows lower prevalence rates.

6. PLOS authors have the option to publish the peer review history of their article (what does this mean?). If published, this will include your full peer review and any attached files.

Reviewer #1: No

Reviewer #2: **Yes: **Thomas M Crea

---

## [Author Response · Author response to Decision Letter 0]

3 Oct 2020

Dear John Schieffelin,

Re: Response to Reviewer’s Comments

We appreciate the reviewer’s suggestions and comments on our manuscript and have carefully gone through the details and made sure the revised manuscript fully addresses the editor’s and reviewer’s comments.

Below we have outlined our responses in the attached Response to Review Comments.

The authors thank the reviewer for bringing these matters to our attention, we hope we have successfully addressed them.

Furthermore as a result of those comments we have substantially revised the whole manuscript.

Response to Review Comments 

Specifically, the Reviewer made the following comments and we have made the following changes to the manuscript: (full details of the Reviewer’s comments are shown below):

Reviewer 1

Reviewer Comment 1.

First, this should not be titled and described as a study of PTSD symptoms. Anxiety and depression are not specific PTSD symptoms, although some PTSD symptoms are similar to them (e.g.,. hyperarousal, negative mood). PTSD may involve insomnia but not all insomnia is trauma related. It may be that experiencing Ebol;a was traumatizing, but without evidence linking the symptoms to Ebola-related experiences these cannot be assumed to be PTSD, they may be due to other stressors or pre-existing conditions, for example. The study should be described as assessing anxiety, depression, and insomnia symptoms, and the possible link to traumatization can be noted but not assumed. The lengthy and very basic description of PTSD should be deleted or greatly shortened.

Response

We thank the reviewer for this observation:

We have accepted the suggestion and adopted a new title which now reads;

“Assessing anxiety, depression, and insomnia symptoms among Ebola survivors in Africa: a meta-analysis”.

We have also accepted the recommendation to delete the definition of post-traumatic stress disorder (PTSD). This has been deleted from the document.

Reviewer Comment 2

“Second, how does this study add new information beyond that reported in other reviews? Is it the first to bring together these or similar studies? IF not, what new information does it provide other than showing that prevalence estimates of the target symptoms are generally consistent across nationalities?”

Response

We thank the reviewer for this observation:

Few studies (2) have qualitatively assessed the psychological and neuropsychological consequences as well as lived experiences of Ebola survivors. To the authors’ knowledge, no study has quantitatively estimated the prevalence of anxiety, depressive symptoms, and insomnia among Ebola survivors. Accurate estimates are required for informed policy planning. This is stated in our manuscript on page 3 line 10-18 as;

“Currently, the majority of existing reviews have been qualitative assessments of psychological and neuropsychological consequences, lived experiences and coping strategies of Ebola disease among survivors [23,24]. Also, there appears to be non-existing meta-analysis evidence on the prevalence of anxiety, depression and insomnia symptoms as mental health burden among Ebola survivors in the affected countries in Africa”. Thus, this meta-analysis presents a quantitative estimate of the prevalence of the above symptoms among EVD survivors. Providing this evidence could support clinical and policy efforts to improve post-health crisis interventions and resources allocation to mitigate the mental health impacts for survivors, families, and frontline workers….”

Reviewer comment 3

“The absence of any comparison to estimates of the target conditions is understandable given that most of the studies did not include non-Ebola control groups. However, whether the prevalence estimates are clinically significant needs to be justified by contrasting the pooled estimates with prevalence data from community epidemiological studies. The prevalence of estimates for symptoms of anxiety (14%; 99% CI: 0.05–0.30), depression (15%; 99% CI: 0.11–0.21), and insomnia (22%; 99% CI: 0.13–0.36) are not clearly higher than estimates from other population surveys, making the authors' conclusion that "a significant amount of EVD survivors are struggling with common PTSD symptoms" unjustified”.

We acknowledge that, prevalence estimates in the general population surveys may be higher than those among only Ebola virus disease survivors. We also note that our restriction of this study to only survivors of Ebola diseases might have led to underreporting compared to the general population. Our findings are similar to a recently published paper. We have duly stated that in our manuscript which now reads;

“Our results are slightly higher than what was reported in a recently published three-country study that found a 10.7%, 9.9% and 4.2% prevalence in anxiety among Ebola survivors in Sierra Leone, Liberia and Guinea respectively [60]. It is interesting to note that, studies that are restricted to only Ebola survivors turn to report lower estimates of anxiety, depression and insomnia compared to those conducted among the general population. In contrast to what we found, previous studies that reported on the mental health consequence of post-disaster occurrences in Sierra Leone showed a higher anxiety prevalence of 48% (95% CI: 46.8% to 50.0%) among the general population [53, 54]. Although the reasons for the low prevalence estimates in these symptoms specifically among Ebola survivors are unclear, at the population level, researchers attributed the higher prevalence of people with mental health problems to region of residence, experiences with Ebola and perceived Ebola threat or knowing someone quarantined for Ebola [53, 54]”.

Reviewer comment 4

“Additionally, information is needed about how each of the conditions was operationalized in the various studies. Some studies used questionnaires: how were "anxiety" "depression" and "insomnia" determined to be present and clinically significant in the different studies? If symptom levels with a cut-off was used, was this based on relevant validation evidence? Other studies relied on medical reports: what was reported (diagnoses? impressions?) and what evidence was provided that the report yielded a valid classification of clinically significant symptoms?”

Response

We also note this observation.

In Table 1 of our study, we provide a list of the assessment instruments that were used by each study to measure the health outcomes of interest that our paper focused on. Several of the studies relied on standardized questionnaires and clinical assessments to provide diagnosis whereas others used medical reports. Comprehensive details of these measures are provided in the methods section of each paper we used in this meta-analysis. However, some of the papers used did not provide details of their clinical diagnostic criteria and the clinical classification standards used to conduct their assessments of the clinical conditions we focused on. Therefore, our inability to access this information limits our ability to comment on this comment as we were unable to access this information from the authors we contacted. We also recognize that this is a further limitation of our study.

Reviewer comment 5 

The authors state that there was no publication bias, but how they determined this needs to be explained carefully.

Response

We thank the reviewer for the observations. 

Indeed, we recognize the need to ensure our estimates reflect accurately on the full spectrum of Ebola studies regarding the three conditions we focused on. Therefore, we have provided a full description of how we assessed publication bias in the data analysis and synthesis section of our paper. We relied on several statistical techniques to determine publication bias including funnel plots based on standard errors and the Eagger unweighted regression test (which is deemed suitable for meta-analysis using a small number of studies, less than 25 studies). Related to this, we also used the Leave-one-out diagnostic test to further confirm the effect of each study on the overall mean and estimate in order to control for overly influential studies in the meta-analysis as multiple studies conducted within the same country could have used overlapping samples.

Reviewer comment 6 

Also, a major possible bias is the selection of participants. How did the studies demonstrate that their samples were representative of the larger group of Ebola survivors in their setting? If the samples are not clearly representative the findings are of limited value.

Response

We thank the reviewer for the observation.

More information on the details of each article included in this study can be found in Table 1, of our paper. It was our general observation that these studies each used a fairly representative sample size from the general Ebola survivor population within each respective study country. Several of the papers provided comprehensive summary statistics of the total Ebola survivors’ population in the area studied and the total number of that population used in their given studies. This information can be found in the methods section of each study. However, as our study mainly focused on the three conditions (i.e. anxiety, insomnia, and depression), we relied on the section of the total sample size for whom complete data on proportions was provided relating to these three conditions.

Reviewer comment 7 

Were there no data on the severity of Ebola illness? This may distinguish individuals who have these symptoms from others who do not. If this was not reported it should be noted as a limitation.

Response

We acknowledge this limitation in our manuscript in page 15, lines 26-28 and it now reads;

“Thirdly, there were no data on the severity of Ebola illness in all the papers assessed. Thus, our study could not distinguish between individuals who have these symptoms from others who do not….”

Reviewer comment 8 

Was gender not reported? Comparing men vs. women would add to the usefulness of the results.

Response

We thank the reviewer for the observation.

Gender was not consistently reported across board in the papers analysed. We could not do sub-analysis by gender, because it will not give a fair representation of the papers and may affect the reliability of our results. Our initial goal was to include sub-analysis to the paper based on different demographic variables. However, there was limited data provided on the several demographic variables related to anxiety, depression, and insomnia, and we were unable to obtain the raw data from the authors of the included studies that we reached out to. In addition, the small number of studies under each analysis stratum would not permit any meaningful sub-group analysis. Evidence of these emails can be provided should there be a request. This has been stated in our manuscript in page 15, lines 22,23 as a limitation of our study.

Reviewer comment 9

Statements that describe the study or its findings in congratulatory manner should be deleted (e.g., "exciting" findings).

Response

We thank the reviewer for the observation

The word “exciting” has been deleted and the sentence now reads;

“Generally, our meta-analysis revealed findings regarding the presence of symptoms of anxiety, depression, and insomnia among Ebola virus disease survivors….”

Reviewer 2

Reviewer comment 1

I would avoid beginning the abstract with a statement of the study’s timeliness. It would be better to lead with, “During health disaster events….”

Response

The authors thank the reviewer for the observation

The background section of the abstract has been reworded and it now reads;

“During health disaster events such as the current devastating havoc being inflicted on countries globally by the SARS-CoV-2 pandemic, mental health problems among survivors and frontline workers are likely….”

Reviewer comment 2

I am not that familiar with methods in meta analyses, but I am accustomed to seeing effect sizes reported. In addition to prevalence estimates, is it possible to report effect sizes?

Response

We appreciate the reviewer’s recommendation and have adopted it and added effect estimates to our results section in the abstract which now reads;

“Effect estimates ranging from (0.13; 99% CI: 0.05, 0.28) through to (0.11; 99% CI: 0.05–0.22), (0.15; 99% CI: 0.09–0.25) through to (0.13; 99% CI: 0.08–0.21) and (0.23; 99% CI: 0.11–0.41) to (0.23; 99% CI: 0.11–0.41) were respectively reported for anxiety, depression and insomnia symptoms…”

Reviewer comment 3

The literature reviewed in this section is thorough and highlights the problems of mental health care particularly in LMICs. The section could be better organized to make it easier for the reader to digest, either through (a) more effective use of introductory sentences in each paragraph, or (b) the use of headings and subheadings.

Response

We have accepted the recommendation of the reviewer and used introductory sentences such as in addition, also, however, therefore, currently etc. to link various paragraphs.

Reviewer comment 4

The word “unfortunately” is used too much – in particular, see the beginning sentences on p. 10 and p. 11.

Response

 We appreciate the reviewer’s observation

The usage of the word “unfortunately” has been reduced to two and replaced with words such as “however”, “therefore” etc.

Reviewer comment 5

This section is clear and succinct. For study inclusion and exclusion criteria, it would be helpful to know the number of studies included and excluded per criterion – if possible, within the flow chart depicted on p. 13.

Response

We followed standard recommended practice of inclusion and exclusion criteria in our article screening process using the flow chart. It should however be noted that the number of studies included and excluded per criterion in this study can be found under search results in page 5. 

Reviewer comment 6

The information provided in the first paragraph is redundant with information already provided in the Methods section.

Response

We thank the reviewer for this observation. 

We have deleted the redundant information from the first paragraph of the results section and the paragraph now reads;

“A total of 13 remaining articles were included in the meta-analysis [40-52]. Table 1 shows a detailed summary of the study attributes and data on the characteristics of the reviewed articles. Data were extracted based on the following features: country, design, sample, sex, age, follow-up period, presence of anxiety, depression, and insomnia. We considered the quality of the studies reviewed as high as there was no observed evidence of any publication bias….” 

Reviewer comment 7

The first paragraph basically repeats earlier information – I would advise starting this section with the main contribution of this study to existing literature.

Response

We again thank the reviewer for this observation. 

We have deleted the redundant information from the first paragraph of the discussion section and the paragraph now reads;

“The objective of this study was to provide pooled prevalence estimates of some common mental health problems amid health disaster events in West-Africa. Generally, our meta-analysis revealed findings regarding the presence of symptoms of anxiety, depression, and insomnia among Ebola virus disease survivors….”

Reviewer comment 8

On p. 22, replace with word “exciting” with a word more appropriate to the situation: “Generally, our meta-analysis revealed exciting findings….”

We essentially dealt with this point under response 9 to Reviewer#1. We repeat it here 

We thank the reviewer for the observation

The word “exciting” has been deleted and the sentence now reads;

“Generally, our meta-analysis revealed findings regarding the presence of symptoms of anxiety, depression, and insomnia among Ebola virus disease survivors…”

Reviewer comment 9

Also, on p. 22, the authors report a total of 3,042 EVD survivors included in the study. Are the authors sure that the various studies all used different samples? Or could there be overlap across studies in terms of participants?

Response

We are grateful to the reviewer for this observation. Indeed, as a meta-analysis study, we were limited to available studies that were conducted in specific countries where the Ebola virus was peculiar to. In this regard, multiple studies using different methods were conducted on the Ebola virus within some countries by the different researchers. Although the authors admit that there might be an overlap across studies in terms of participants, we undertook several statistical measures to control for some amount of this overlap including considering only raw frequencies of reported symptoms of anxiety, depression and insomnia to estimate their prevalence. Also, as heteroscedasticity is a major concern when samples are assumed to be related, we undertook several statistical measures including using the Q-statistic (X2) and SerSimonian and Laird I2 statistic were used to determine heterogeneity across the various studies used. In addition, the Leave-one-out statistical technique was used to identify the effect of each study on the overall mean and estimate for the observed summary proportions. This allowed us to control for outlying studies that could have occurred due to having correlated samples. These techniques are discussed in full in the Data analysis and synthesis section of our study.

Reviewer comment 10

In the Discussion section, please give some thought as to potential mechanisms behind the various prevalence rates presented. For example, why might insomnia have a higher prevalence than anxiety and depression? Also, the prevalence rates in the current study are much lower across the board compared to individual studies. Please go into some detail as to why the meta-analysis shows lower prevalence rates.

We essentially dealt with this point under response 3 to Reviewer#1. We repeat it here 

We acknowledge that, prevalence estimates in the general population surveys may be higher than those among only Ebola virus disease survivors. We also note that our restriction of this study to only survivors of Ebola diseases might have led to underreporting compared to the general population. Our findings are similar to a recently published paper. We have duly stated that in our manuscript which now reads;

Our results are slightly higher than what was reported in a recently published three-country study that found a 10.7%, 9.9% and 4.2% prevalence in anxiety among Ebola survivors in Sierra Leone, Liberia and Guinea respectively [60]. It is interesting to note that, studies that are restricted to only Ebola survivors turn to report lower estimates of anxiety, depression and insomnia compared to those conducted among the general population. In contrast to what we found, previous studies that reported on the mental health consequence of post-disaster occurrences in Sierra Leone showed a higher anxiety prevalence of 48% (95% CI: 46.8% to 50.0%) among the general population [53, 54]. Although the reasons for the low prevalence estimates in these symptoms specifically among Ebola survivors are unclear, at the population level, researchers attributed the higher population of prevalence the mental health problems to region of residence, experiences with Ebola and perceived Ebola threat or knowing someone quarantined for Ebola [53, 54].

The authors thank the reviewers for bringing these matters to our attention, we hope we have successfully addressed them.

---

## [Decision Letter · Decision Letter 1]

11 Nov 2020

PONE-D-20-14937R1

Assessing anxiety, depression and insomnia symptoms among Ebola survivors in Africa: a meta-analysis

PLOS ONE

Dear Dr. Mengesha,

Thank you for submitting your manuscript to PLOS ONE. After careful consideration, we feel that it has merit but does not fully meet PLOS ONE’s publication criteria as it currently stands. Therefore, we invite you to submit a revised version of the manuscript that addresses the points raised during the review process.

We look forward to receiving your revised manuscript.

Kind regards,

John Schieffelin, MD

Academic Editor

PLOS ONE

Reviewers' comments:

Reviewer's Responses to Questions

**Comments to the Author**

1. If the authors have adequately addressed your comments raised in a previous round of review and you feel that this manuscript is now acceptable for publication, you may indicate that here to bypass the “Comments to the Author” section, enter your conflict of interest statement in the “Confidential to Editor” section, and submit your "Accept" recommendation.

Reviewer #1: All comments have been addressed

Reviewer #2: All comments have been addressed

2. Is the manuscript technically sound, and do the data support the conclusions?

Reviewer #1: Yes

Reviewer #2: Yes

3. Has the statistical analysis been performed appropriately and rigorously? 

Reviewer #1: Yes

Reviewer #2: Yes

4. Have the authors made all data underlying the findings in their manuscript fully available?

Reviewer #1: Yes

Reviewer #2: Yes

5. Is the manuscript presented in an intelligible fashion and written in standard English?

Reviewer #1: Yes

Reviewer #2: Yes

6. Review Comments to the Author

Reviewer #1: Very thoughtful and responsive revision, the paper now is a solid contribution to the field. I have a few additional revisions to suggest:

1. Table 1: delete "(years)" from the header (several entries are in months)

2. Results: Explain briefly how the studies confirmed the past diagnosis of Ebola (apparently this was done by medical records in some cases and self-report in others) and consider examining whether the findings differ based on whether the health outcomes were assessed by medical records versus self-report (it appears from the sensitivity analyses that this did not make a difference, but it would be good to be explicit about that).

3. I don't think that the lower prevalence estimates in this meta-analysis are due to a "fewer number of studies used" -- a more likely reason is that this meta-analysis included only studies that identified clinically-significant anxiety, depression, and insomnia symptoms where the prior reviews may have included estimates based on symptoms that were mild as well.

4. Consider the possibility that unlike surviving a mass disaster that involved intentional harm (e.g., war) or that destroyed entire communities (like natural or humanmade disasters), surviving Ebola may not cause a chronic emotional injury (except for persons who lost loved ones or experienced extreme suffering and feared they would die), and those who do survive may be relatively physically and emotionally resilient. This could explain the relatively normative prevalence estimates.

5. The statement that this "study reveals, survivors ... are at increased risk of symptoms ..." is not accurate. The findings suggest that EVD survivors may experience problems with anxiety, depression, and insomnia, but not that they are at increased risk because the prevalence estimates are not clearly higher than those for general community samples in Africa (or other parts of the world). The findings do suggest that research is needed to determine what risk factors (e.g., severity of the disease, loss of loved ones, fear of dying) distinguish EVD survivors who have these problems from those who do not. If risk factors are identified, then prevention and treatment can be targeted to address them, which is more feasible and cost-effective than a universal program to prevent mental health problems for all EVD survivors (most of whom apparently are not experiencing problematic anxiety, depression, or insomnia).

Reviewer #2: All of my comments have been adequately addressed. Thanks to the authors for their responsiveness. The article will make a positive contribution to the literature on Ebola and mental health.

7. PLOS authors have the option to publish the peer review history of their article (what does this mean?). If published, this will include your full peer review and any attached files.

Reviewer #1: No

Reviewer #2: **Yes: **Thomas M. Crea

---

## [Author Response · Author response to Decision Letter 1]

10 Dec 2020

Dear John Schieffelin,

Re: Response to Reviewer’s Comments

We appreciate the reviewer’s suggestions and comments on our manuscript and have carefully gone through the details and made sure the revised manuscript fully addresses the editor’s and reviewer’s comments.

Below we have outlined our responses in the attached Response to Review Comments.

The authors thank the reviewer for bringing these matters to our attention, we hope we have successfully addressed them.

Furthermore as a result of those comments we have substantially revised the whole manuscript.

Response to Review Comments 

Specifically, the Reviewer made the following comments and we have made the following changes to the manuscript: (full details of the Reviewer’s comments are shown below):

Reviewer 1

Opening comment

Very thoughtful and responsive revision, the paper now is a solid contribution to the field. I have a few additional revisions to suggest

Response

The authors are grateful to the reviewer for the kind words and his/her thought-provoking comments that have contributed towards improving the quality of this manuscript. 

Reviewer Comment 1.

Table 1: delete "(years)" from the header (several entries are in months)

Response

We thank the reviewer for this observation:

We have accepted the suggestion and deleted years from the header in table 1. 

Reviewer Comment 2

“Results: Explain briefly how the studies confirmed the past diagnosis of Ebola (apparently this was done by medical records in some cases and self-report in others) and consider examining whether the findings differ based on whether the health outcomes were assessed by medical records versus self-report (it appears from the sensitivity analyses that this did not make a difference, but it would be good to be explicit about that)”.

Response

We thank the reviewer for this observation:

“We agree with the suggestion to explicitly confirm if any differences existed in the findings based on the assessment of health outcome of interest. However, while we acknowledge strongly that there is a need for further sub-analysis, we would like to kindly mention that all the studies included in the meta-analysis confirmed an Ebola diagnosis using medically approved clinical tests which were not included in this meta-analysis as this topic was beyond the scope of the paper. We also note that these self-reports and medical records were mainly used to examine patients’ mental health states including the presence or absence of insomnia, depression, and anxiety. Moreover, as mentioned in our earlier response, we were unable to conduct further sub-analysis on covariates including the assessment of health outcomes because of the small sample size of articles included in the meta-analysis. We have now included this as a potential limitation in our paper and a recommendation for future studies on this or similar topics.”

Reviewer comment 3

“I don't think that the lower prevalence estimates in this meta-analysis are due to a "fewer number of studies used" -- a more likely reason is that this meta-analysis included only studies that identified clinically-significant anxiety, depression, and insomnia symptoms where the prior reviews may have included estimates based on symptoms that were mild as well”.

Response

We accept the recommendation of the reviewer and have incorporated the comment into the manuscript and it now reads;

“We believe that a more likely reason for the differences in the estimated proportions is that this meta-analysis included only studies that identified clinically-significant anxiety, depression, and insomnia symptoms where the prior reviews may have included estimates based on symptoms that were mild as well”.

Reviewer comment 4

“Consider the possibility that unlike surviving a mass disaster that involved intentional harm (e.g., war) or that destroyed entire communities (like natural or humanmade disasters), surviving Ebola may not cause a chronic emotional injury (except for persons who lost loved ones or experienced extreme suffering and feared they would die), and those who do survive may be relatively physically and emotionally resilient. This could explain the relatively normative prevalence estimates”

Response

We accept the recommendation of the reviewer and have incorporated the comment into the manuscript and it now reads;

“A possible explanation to our findings is that unlike surviving a mass disaster that involved intentional harm (e.g., war) or that destroyed entire communities (like natural or humanmade disasters), surviving Ebola may not cause a chronic emotional injury (except for persons who lost loved ones or experienced extreme suffering and feared they would die), and those who do survive may be relatively physically and emotionally resilient.”

Reviewer comment 5 

“The statement that this "study reveals, survivors ... are at increased risk of symptoms ..." is not accurate. The findings suggest that EVD survivors may experience problems with anxiety, depression, and insomnia, but not that they are at increased risk because the prevalence estimates are not clearly higher than those for general community samples in Africa (or other parts of the world). The findings do suggest that research is needed to determine what risk factors (e.g., severity of the disease, loss of loved ones, fear of dying) distinguish EVD survivors who have these problems from those who do not. If risk factors are identified, then prevention and treatment can be targeted to address them, which is more feasible and cost-effective than a universal program to prevent mental health problems for all EVD survivors (most of whom apparently are not experiencing problematic anxiety, depression, or insomnia)”.

Response

We thank the reviewer for the suggestion. This has been incorporated into the manuscript and it now reads;

“Accordingly, our findings suggest that EVD survivors may experience problems with anxiety, depression, and insomnia. The findings also suggest that research is needed to determine what risk factors (e.g., severity of the disease, loss of loved ones, fear of dying) distinguish EVD survivors who have these problems from those who do not. If such risk factors are identified, then prevention and treatment can be targeted to address them, which is more feasible and cost-effective than a universal program to prevent mental health problems for all EVD survivors (most of whom apparently are not experiencing problematic anxiety, depression, or insomnia). Such coordinated targeted interventions will demand comprehensive collaboration between governments, health professionals, donors, civil society, communities, and patients and their families to cater for vulnerable populations such as EVD survivors as recommended by the WHO Mental Health Gap Action Programme [59]”.

Reviewer 2

Reviewer comment 1

All of my comments have been adequately addressed. Thanks to the authors for their responsiveness. The article will make a positive contribution to the literature on Ebola and mental health

Response

The authors are grateful to the reviewer for the kind words and thankful for him/her agreeing to review our manuscript that will contribute to the literature on Ebola and mental health

---

## [Decision Letter · Decision Letter 2]

21 Jan 2021

Assessing anxiety, depression and insomnia symptoms among Ebola survivors in Africa: a meta-analysis

PONE-D-20-14937R2

Dear Dr. Mengesha,

We’re pleased to inform you that your manuscript has been judged scientifically suitable for publication and will be formally accepted for publication once it meets all outstanding technical requirements.

Kind regards,

John Schieffelin, MD

Academic Editor

PLOS ONE

Additional Editor Comments (optional):

Reviewers' comments:

Reviewer's Responses to Questions

**Comments to the Author**

1. If the authors have adequately addressed your comments raised in a previous round of review and you feel that this manuscript is now acceptable for publication, you may indicate that here to bypass the “Comments to the Author” section, enter your conflict of interest statement in the “Confidential to Editor” section, and submit your "Accept" recommendation.

Reviewer #1: All comments have been addressed

Reviewer #2: All comments have been addressed

2. Is the manuscript technically sound, and do the data support the conclusions?

Reviewer #1: Yes

Reviewer #2: Yes

3. Has the statistical analysis been performed appropriately and rigorously? 

Reviewer #1: Yes

Reviewer #2: Yes

4. Have the authors made all data underlying the findings in their manuscript fully available?

Reviewer #1: Yes

Reviewer #2: (No Response)

5. Is the manuscript presented in an intelligible fashion and written in standard English?

Reviewer #1: Yes

Reviewer #2: (No Response)

6. Review Comments to the Author

Reviewer #1: Excellent revision and solid contribution to the research literature on an important topic.

Reviewer #2: All of my comments have been adequately addressed. Thanks to the authors for their

responsiveness. The article will make a positive contribution to the literature on Ebola

and mental health.

7. PLOS authors have the option to publish the peer review history of their article (what does this mean?). If published, this will include your full peer review and any attached files.

Reviewer #1: No

Reviewer #2: No

---

## [Editor Report · Acceptance letter]

26 Jan 2021

PONE-D-20-14937R2 

Assessing anxiety, depression and insomnia symptoms among Ebola survivors in Africa: a meta-analysis 

Dear Dr. Mengesha:

I'm pleased to inform you that your manuscript has been deemed suitable for publication in PLOS ONE. Congratulations! Your manuscript is now with our production department. 

Kind regards, 

on behalf of

Dr, John Schieffelin 

Academic Editor

PLOS ONE